# Progress and Viewpoints of Multifunctional Composite Nanomaterials for Glioblastoma Theranostics

**DOI:** 10.3390/pharmaceutics14020456

**Published:** 2022-02-21

**Authors:** Ming-Hsien Chan, Wen-Tse Huang, Aishwarya Satpathy, Ting-Yi Su, Michael Hsiao, Ru-Shi Liu

**Affiliations:** 1Department of Chemistry, National Taiwan University, Taipei 106, Taiwan; ahsien0718@gate.sinica.edu.tw (M.-H.C.); d09223104@ntu.edu.tw (W.-T.H.); d09223109@ntu.edu.tw (A.S.); b06203072@ntu.edu.tw (T.-Y.S.); 2Genomics Research Center, Academia Sinica, Taipei 115, Taiwan; 3Department of Biochemistry, College of Medicine, Kaohsiung Medical University, Kaohsiung 807, Taiwan

**Keywords:** glioblastoma, magnetic resonance imaging nanoparticles, near-infrared probes, near-infrared phototherapy, FDA-proved drugs

## Abstract

The most common malignant tumor of the brain is glioblastoma multiforme (GBM) in adults. Many patients die shortly after diagnosis, and only 6% of patients survive more than 5 years. Moreover, the current average survival of malignant brain tumors is only about 15 months, and the recurrence rate within 2 years is almost 100%. Brain diseases are complicated to treat. The reason for this is that drugs are challenging to deliver to the brain because there is a blood–brain barrier (BBB) protection mechanism in the brain, which only allows water, oxygen, and blood sugar to enter the brain through blood vessels. Other chemicals cannot enter the brain due to their large size or are considered harmful substances. As a result, the efficacy of drugs for treating brain diseases is only about 30%, which cannot satisfy treatment expectations. Therefore, researchers have designed many types of nanoparticles and nanocomposites to fight against the most common malignant tumors in the brain, and they have been successful in animal experiments. This review will discuss the application of various nanocomposites in diagnosing and treating GBM. The topics include (1) the efficient and long-term tracking of brain images (magnetic resonance imaging, MRI, and near-infrared light (NIR)); (2) breaking through BBB for drug delivery; and (3) natural and chemical drugs equipped with nanomaterials. These multifunctional nanoparticles can overcome current difficulties and achieve progressive GBM treatment and diagnosis results.

## 1. Introduction of Brain Glioblastoma

Glioblastoma (GBM) is one of the most common and aggressive brain cancers. The signs and symptoms of glioblastoma are initially nonspecific. Patients may experience headaches, personality changes, nausea, and stroke-like symptoms. The symptoms usually worsen quickly and may develop into unconsciousness [1]. The diagnosis of glioblastoma is usually comprehensively judged by computer tomography, MRI, and biopsy. The treatment of GBM is generally chemotherapy and radiation therapy after surgery [2]. Temozolomide (TMZ) drugs are often used as chemotherapy for glioblastoma [3]. High-dose steroids can be used to help reduce swelling and symptoms. It is unclear whether trying to eradicate the tumor or removing most of the cancer is more helpful to the patient [4]. Glioblastoma usually recurs under complete treatment. After diagnosis, the typical survival period is 12–15 months, and less than 3–7% of people survive for more than five years. Without treatment, the survival period is usually only 3 months, which shows that GBM is a very deadly cancer. GBM is difficult to treat because the brain tissue is intertwined under the blood–brain barrier (BBB), composed of endothelial cells, astrocytes, and the basement membrane [5]. This “barrier” can selectively prevent certain substances from being removed from the blood into the brain, allowing only small, necessary molecules such as oxygen, amino acids, glucose, and water to pass through and isolating large molecules such as drugs and proteins. The BBB protects the brain from infection by germs and makes intracranial treatment challenging to achieve.

GBM has a thick, irregularly developed tumor shell and necrotic tissue in the tumor [6]. Tumors vary in size, and they are brain masses with irregular edges, tumor infiltration, and necrotic tissue. Under the general detection system, the shape of the tumor tissue cannot be effectively observed. At this time, MRI is generally relied upon to detect GBM [7]. MRI is sensitive and can easily find small tumors that cannot be seen by CT [8]. In addition to MRI diagnosis, optical analysis is also considered to be another noninvasive diagnostic method [9]. However, due to the limitation of the depth of light, the current light source that can penetrate the brain skull tissue and perform treatment and diagnosis is near-infrared. Since it is low-energy and high-wavelength, NIR can pass through around 3–10 cm of brain tissue to enter the GBM for optical detection. In order to achieve high-efficiency detection of GBM through MRI and NIR diagnosis, current research points out that multifunctional nanomaterials are highly influential diagnostic platforms that can improve the signal-to-noise ratio of the diagnostic process. In addition, nanomaterials have an easily modified surface structure that allows them to penetrate the BBB for targeted therapy after conjugating with specific targeting ligands. For instance, by adding a transferrin coating to liposomal nanoparticles, researchers can enable the nanoparticles to smoothly pass through the BBB and avoid normal cells [10]. The nanocomposite that could accurately reach the tumor site to the brain tumor cells were removed from patients and cultured in the laboratory [11]. After that, traditional chemotherapeutic drugs, gold, and special liposomes are combined into new types of nanoparticles, injected into the brain tumor cells in a petri dish, and placed in a radiotherapy environment. The gold releases electrons and destroys cancer cells [12]. Its DNA and overall structure enhance the effect of chemotherapy drugs [13]. This transferrin-coated liposome for drug delivery has a better therapeutic effect than the traditional direct injection of drugs [14]. The brain tumors of experimental mice have been reduced in size, and the survival rate has also been improved. The curative effect is quite apparent in the experimental results. After several days, the cultured tumor cells showed no signs of recurrence, which means that the brain cancer cells have been completely destroyed [15]. Because the new method can avoid some adverse reactions caused by the direct injection of drugs, it can be used to deliver other anticancer drugs, and it can be used to carry other drugs that cannot break through the BBB to treat other brain diseases [16].

Some review articles have discussed the contribution and breakthrough of nanomaterials to GBM treatment [17]. However, many challenges still need to be overcome based on the previous research: first, the percentage of nanomaterials accumulating the tumor must be increased. According to the current research results, it is calculated that if nanomaterials cannot break through the BBB, only about 11% of particles will reach the mouse brain tumor site. This means that the remaining 89% may spread into other organs that we cannot expect. Secondly, the total amount of drugs carried by the nanomaterials needs to be accurately controlled. Even if the targeting ligands have been modified on the surface to make nanomaterials accumulate in the tumor, the premature release of the drugs still needs to be concerned. This review will discuss the application of various nanocomposites in diagnosing and treating GBM. With the continuous advancement of nanotechnologies, current research has made more significant breakthroughs (Figure 1). The topics include 1. the efficient and long-term tracking of brain images (MRI and NIR); 2. passing through BBB for drug delivery; and 3. natural and chemical drugs equipped with nanomaterials. Inorganic/organic nanomaterials exhibit T2-weighted magnetic and light conversion properties, effectively diagnosing GBM. Carrying drugs with different natural and chemical molecules makes this platform more potent for the treatment of GBM. We will demonstrate that these multifunctional nanoparticles can overcome the current difficulties and achieve progressive GBM treatment and diagnosis results.

## 2. Magnetic Resonance Imaging and Therapy of Brain Glioblastoma

Most of the currently researched and developed nano-MRI imaging agents are iron nanomaterials. These MRI agents have been developed many times and used in actual research, such as iron-based nanomaterials for GBM chimeric antigen receptor T cell (CAR-T cell) therapy animal experiments [18]. Multifunctional iron oxide nanoparticle technology can mark GBM through T2-weighted MRI [19]. This type of nano-iron oxide MRI imaging agent even passed the U.S. Food and Drug Administration (FDA) Human Clinical Trial Review (IND) in February 2018 and was approved for Phase II clinical trials, compared to the use of MRI alone for the diagnosis of GBM. The addition of iron-based nanomaterials can increase the accuracy of MRI recognition to almost 100% [20] and improve the contrast of GBM under MRI more than 10 times, which is significantly better than other imaging products on the market. Due to iron-based nanomaterials’ high biocompatibility and low toxicity, their adverse reaction rate is only 3.84%, significantly lower than the average level of 10–15% of other developers. They are considered to be able to assist the medical team in providing a better diagnosis plan for GBM patients in the future [21].

Essentially all MRI nanomaterials suitable for GBM treatment are iron-based nanoparticles. Thus, this section will be divided into two parts to discuss iron-based nanotechnology platforms: “diagnosis” and “therapy”. These studies have entered preclinical development. Iron-based nanoparticles combined with organic matter coating can make the iron-based particles quickly swallowed by immune cells by regulating the hydrophilic and hydrophobic properties of the surface, which can indicate malignant tumors, and the patient has no doubts about heavy metal deposition and nephropathy.

### 2.1. Diagnosis

In MRI examination, the most commonly used imaging agent is the imaging agent containing gadolinium (Gd), but gadolinium ion is a lanthanide heavy-metal element, which is inherently toxic [6]. It must be combined with highly stable compounds to reduce heavy metal residues, and product stability is tricky. Control of gamma ions and their nephrotoxicity and doubts about accumulation in the brain have also received considerable attention [22]. Due to the unique properties of magnetic nanoparticles, such as superparamagnetism, high saturation magnetization, and high effective surface area, they have been used in biomedical fields such as biomedical diagnostic imaging, disease treatment, and biochemical separation [23].

In biomedical diagnostic imaging, it is mainly used as a contrast agent to improve image contrast and as a tracer for the calibration of specific targets; in disease treatment, it can be used as a carrier for drug delivery, or an external magnetic field can be used to generate heat energy from magnetic nanoparticles injected into the body to kill cancer cells, which can alleviate damage to normal cells caused by traditional chemotherapy [24]. Superparamagnetic iron oxide properties provide the carrier with unique capabilities, such as magnetic guidance and MRI (Figure 2a) [25]. Finally, the magnetic drug carrier is modified with lactoferrin (Lf) on the surface so that nanocomposites can cross BBB and effectively increase the efficiency of entering the brain (Figure 2b). In order to improve the permeability of the BBB, Yee et al. first wrapped iron oxide and paclitaxel, an anti-cancer chemotherapy drug [26]. As necrosis progresses and enlarges, the damaged/dying tumor cells may recruit even more neutrophils to infiltrate into necrotic niches (Figure 2c). Finally, necrosis and neutrophil infiltration may form a positive feedback loop to amplify intratumoral necrosis formation to its total capacity [27]. In order to compare BBB-crossing efficiency, Fang et al. was performed in vivo MRI with glioma-bearing rats after injection with magnetic nanocarriers or Lf-magnetic nanocarriers. The Lf receptor is overexpressed in glioma, which enhanced BBB-crossing efficiency after injection of Lf-conjugated magnetic nanocarriers and caused a significant increase in the amount of Lf-magnetic nanocarriers around the vascular region of the brain tumor tissue. (Figure 2d). In turn, the intracellular toxicity of these nanoparticles to neutrophils is reduced. After injecting these neutral ball robots into the blood vessels of mice, an external magnetic field can be used to control and accelerate the swimming direction and speed of these robots in the blood vessels [28]. In addition, the neutrophils will actively pass over them (Figure 2e). The blood–brain barrier moves in the direction of inflammation in the brain. Therefore, compared with traditional drug injection, the robot has two more propulsion forces, significantly improving efficiency and speed.

### 2.2. Therapy

In the past, studies have tried to use magnetic nanoparticles or ultrasonic waves to switch drugs, but this will induce other risks; currently, a group of research teams has developed new nanoparticles that pretend to be amino acids to “deceive” barriers. The drug will be carried to the tumor when it enters the brain and is released. First of all, in the past, scientists have observed the overexpression of LAT1 molecules in a variety of tumor cells. Hence, the research team designed nanoparticles to be recognized and bound by LAT1 molecules [29]. In this manner, nanoparticles are similar to missiles equipped with a global positioning system. After the brain is blocked, the affected area with many LAT1 molecules can be locked, and tumors can be found. In the past, other nanoparticles have been able to penetrate the barrier. However, the researchers pointed out that the previous nanoparticle design was very complicated and sometimes could not be removed smoothly from the brain.

In contrast, the newly designed nanoparticle consisted of only a single compound. The structure is straightforward. Although the application of this research to the clinic is still a long way away, this discovery provides a new direction for the development of nanoparticles that target the specific molecule and deliver the drug to the brain [30]. At present, Xie et al. has carried out CAR-T cell therapy experiments, using the ultra-small superparamagnetic particles of iron oxide to label CAR-T cells for non-invasive monitoring of kinetic infiltration and persistence in GBM (Figure 3a) [31]. The primary key to the success of solid tumor treatment is that these cells need to enter the tumor effectively. Based on cell staining, the results demonstrate that iron oxide can be uptakeken by CAR-T cells (Figure 3b). However, there is currently no clinically relevant technology or product for real-time tracking of immune cells in the body, making it impossible for physicians to know whether the cells have entered the tumor in real-time during the treatment process, which usually must be completed (Figure 3c). All treatment courses can be used to evaluate the effectiveness of treatment. If ineffective, it cannot be remedied, and the golden treatment period is missed. The nano-MRI imaging agent made of nano-iron oxide can be successfully labeled on CAR-T cells and endowed with the function of real-time tracking in the body. The results of animal experiments clearly indicate that CAR-T enters brain tumor cell images, which physicians can use in the future. More accurate prediction of treatment effects or immediate feedback increases the treatment success rate. Chan et al. aimed to use nanobubbles (NB) and iron-platinum (FePt) nanoparticles to bypass the blood–brain barrier and assist in the treatment of GBM (Figure 3d) [32]. This research work embeds the treatment drug in an NB and uses a high-intensity focused ultrasound oscillation to break the nanobubble and create a cavitation impact on the BBB (Figure 3e). Temporary cavitation allows the drug to pass through and reach the brain. They also load FePt nanoparticles into the NB, improving the resolution of MRI images. The MRI images from the results of experiments prove that the FePt@NBs successfully guided the drugs to the GBM, where a high-intensity focused ultrasound oscillation broke NB to generate a cavitation opening on BBB, allowing the drugs to bypass it and enter into the brain to treat the GBM tumor (Figure 3f).

Scientists are constantly developing new treatment tactics for cancer. A therapy called “magnetic hyperthermia” is currently still in the experimental stage [33]. Recently, an Oregon State University team has obtained results of magnetic hyperthermia in mice experiments, confirming that it can significantly inhibit the subcutaneous ovaries [34]. The method of magnetic hyperthermia is to inject magnetic nanoparticles directly into the tumor affected by a syringe and then exposed the particles to an alternating magnetic field (AMF) to heat them to a temperature of about 38 °C. The cancer cells have been tested for safety in clinical trials for prostate and brain cancer patients under thermotherapy. However, magnetic hyperthermia has a limited injection range. Magnetic hyperthermia may be misdiagnosed without specific guidance when faced with tumors that cannot be injected directly with nanoparticles. Therefore, Mahmoudi et al. integrated a review work to commit to developing other delivery methods of nanoparticles to accumulate to include other cancers in the scope of magnetic hyperthermia treatment (Figure 3g) [35].

On the one hand, the heating efficiency of nano particles can be improved, and on the other hand, they can be given in small doses to avoid damaging normal cells in the patient’s body [36]. In mouse experiments, Albarqi et al. confirmed that nanoparticle clusters could significantly inhibit the growth of subcutaneous ovarian tumors, opening another door to cancer treatment (Figure 3h). It is more effective in combination with chemotherapy and radiotherapy (Figure 3i,j) [37].

## 3. Near-Infrared Imaging and Therapy of Brain Glioblastoma

### 3.1. Diagnosis

Near-infrared (NIR) imaging is based on the photoluminescence properties of near-infrared emission nanoprobes. With high penetration and low scattering properties, NIR is suitable for biological imaging to explore and analyze information from tissues and organs [38]. The range of the wavelength can also avoid autofluorescence efficiently. According to its bio-optical properties, NIR can be divided into 700–1000 nm, 1000–1350 nm, and 1550–1700 nm, known as the NIR-I, NIR-II, and NIR-III windows [39]. The development of NIR-I emission nanoprobes has matured and is widely used for bio-imaging, tracking, and distribution in vivo [40]. Compared with the NIR-I window, nanoprobes in the NIR-II window can be penetrated more deeply into the living tissue because they present deeper absorption and lower scattering of biological tissues. The transmissive ability of NIR-II and III windows shows a high signal-to-noise ratio, which is conducive to the development of brain imaging [41]. NIR nanoprobes have various types used in glioblastoma diagnosis and therapy recently. They can be classified into organic dye, lanthanide-doped nanoparticles, quantum dots, nanophosphors, and polymer dots, as shown in Table 1.

#### 3.1.1. Organic Dyes

Conventional NIR organic dyes as small molecules need to be improved due to self-quenching, an insufficient Stokes shift between excitation and emission, and degradation in the photo- and bio-environment, such as indocyanine green (ICG), IR700, IR780, IR800, IR825, and chlorin e6 (Ce6) [42,43,44,45,46,47,48,49,50,51,52]. Cetuximab-IRDye800 was used in clinical surgery for Image-Guided Surgery using Invisible Near-Infrared Light to guide the doctor with NIR imaging. The new tumors from patients were cut and measured according to the fluorescence intensity under the Pearl Impulse imaging platform in Figure 4a [48]. However, crossing the BBB by small molecules alone is difficult. The NIR organic dyes can quickly enter the tumor site with the nanoparticle’s assistance. The NIR-II organic dye, IR-FE (emission center: 1040 nm), was recently encapsulated by spherical nucleic acids and it modified the aptamer to be transported to the glioblastoma tumor site. The scheme and brain vessel images of IR-FE are shown in Figure 4b. NIR-II imaging revealed IR-FE’s bright emission and deep penetration with the clear cerebral angiogram under the 808 nm laser [64].

#### 3.1.2. Lanthanide-Doped Nanoparticles

The emission pathway of lanthanide-doped nanoparticles is unique in that it can be presented by upconversion (NIR to visible) and downconversion (NIR to NIR) processes [80]. The upconversion mechanism is based on the anti-Stokes principle, which converts energy into short-wavelength light by absorbing long-wavelength light. Both 980 and 808 nm lasers are familiar excitation sources for deep penetration. The core-shell Tm-doped nanoprobe (NaYF_4_:20%Yb/2%Tm/15%Gd@NaGdF_4_) was a kind of upconversion system used for the intracranial glioblastoma imaging with Angiopep-2 targeting [81]. An average particle size of 17–19 nm was estimated. The signal of the Tm-doped nanoprobe (excitation: 980 nm; emission: 800 nm) from the glioblastoma-bearing brain can be observed in the fluorescence image. By using co-doped strategies with different lanthanide elements, the signal can be enhanced by multiple peaks in the NIR region, such as Er- and Tm-doped nanoparticles (Er: 658 nm; Tm: 795 nm). Moreover, the Er-doped nanoparticle with downconversion process was reported to provide NIR-II emission (1525 nm) for through-skull targeted imaging and imaging-guided surgery of the orthotopic glioma [69]. The scheme of energy transfer from dopant to dopant is shown in Figure 4c. The quantum efficiency of the downconversion is higher than the upconversion system because of amount of energy loss during energy transportation [82]. Significant enhancement was achieved on the Er-doped nanoparticle by 675 folds.

#### 3.1.3. Quantum Dots

The average size of 2–10 nm can be distinguished for most quantum dots, similar to carbon quantum dots, graphene quantum dots, and heavy-metal-free quantum dots. An ideal quantum dot, silver chalcogenides, is a narrow-bandgap material with low toxicity widely used in near-infrared bioimaging [70]. The near-infrared emission can be tuned by the different ratios of silver and chalcogenides. The emission of Ag_2_S quantum dots was located at 1058 nm (NIR-II) from the U87 MG cell line under the irradiation of 658 nm laser diodes [71]. The modified Ag_2_S quantum dots possessed a higher signal intensity in targeted cellular imaging. Graphene quantum dots (4–5 nm) also demonstrated NIR-II emissions at 1000 nm through dual-doped nitrogen and boron [74]. The local distortion and vacancy defects caused the emission peak to red-shift to the NIR-II window by doping with nitrogen and boron. The graphene quantum dots were injected intravenously into the C6 tumor-bearing mouse. The bright NIR-II imaging of the tumor site was acquired using an 808 nm laser.

#### 3.1.4. Nanophosphors

The persistent luminescent property of nanophosphors can be utilized as tracking agents or contrast agents. Persistent luminescence with a long afterglow time and renewability provides another method to avoid autofluorescence. Chromium (Cr) is the common element possessing persistent luminescence through the generation of electron traps. ZnGa_2_O_4_ doped with Cr^3+^ and Sn^4+^ in mesoporous silica nanoparticles presented a longer luminescence at 700 nm combined with the chemotherapy drug for glioblastoma therapy [75]. The nanophosphor modified AS1411 for targeting and linking with nanobubbles. After being injected intravenously into the glioblastoma mouse, NIR nanophosphor accumulated in the tumor site and emitted NIR emission continuously for up to 8 h (Figure 4d).

#### 3.1.5. Polymer Dots

Various types of aggregation-induced emission (AIE) nanoparticles possessing rotor structures have different optical characteristics, such as molecular state (non-emissive) and aggregate state (emissive). A quantity of 1550 nm AIE nanoparticles was utilized to monitor their biodistribution and accumulation in the orthotopic glioma model. After AIE nanoparticle injection, cerebral vasculature and blood vessel imaging can be identified through the skull and scalp.

According to the optical properties of biological tissues, the near-infrared window has lower absorption and scattering, which provides deeper penetration and may result in better brain imaging. However, the luminescent nanoparticles still have been successfully FDA-approved yet. Based on the light pathway from nanoparticles in the brain, nanoparticles excited from NIR light and emitting NIR light are the best solutions. Here, each nanoparticle listed above has different optical properties that can be used independently.

### 3.2. Phototherapy and Other Therapies

In order to address challenges of theranostic strategies of glioblastoma, precise diagnosis and treatments have been developed with significant progress in recent years, such as magnetic hyperthermia, gene therapy, high-intensity focused ultrasound treatment, immunotherapy, and phototherapy [83]. Near-infrared light with high penetration is widely used in phototherapy, such as photothermal and photodynamic therapies. Nanoagents can be activated with the near-infrared laser source [84]. The lower energy distribution and higher penetration of near-infrared laser sources are utilized to activate the nanoagent in the deep brain without damaging normal cells.

The photosensitizer can be excited by an external light source from the ground state (S_0_) to the excited state (S_1_). Subsequently, the energy will return to the ground state via three different pathways with different applications [85]: (1) fluorescence imaging: photon emitting with Stokes shift; (2) persistent luminescence and photodynamic therapy: intersystem crossing from singlet to triplet state with phosphorescence and free radicals or single oxygen; (3) photoacoustic imaging and photothermal therapy: nonradiative relaxation with heat, as shown in Figure 5a. The properties in these three pathways are competitive in the nanomaterials. According to the applications, near-infrared photosensitizers are classified into inorganic (gold nanomaterials, two-dimensional materials, metal oxide materials, and quantum dots) and organic (polymers) types [86].

Near-infrared phototherapy depends on the thermal effect or ROS from photo-sensitizer to kill targeted brain tumors. In order to design nanoplatforms, the absorbance of photosensitizer and nanoparticle luminescence need to be familiar. The energy contribution of the photosensitizer comes from the external light source or the used luminescent nanoparticles. Notably, the absorbance of photosensitizer and luminescence of nanoparticles need to overlap. Assisted with luminescent nanoparticles, treatments and monitoring can be achieved simultaneously. The following paragraph will introduce photothermal and photodynamic therapy in detail.

#### 3.2.1. Photothermal Therapy

Photothermal therapy refers to the effect of a light-driven photosensitizer in producing high temperatures to kill tumors. Vascular nerve receptors have poor sensitivity to temperature. Therefore, raising the ambient temperature will cause varying degrees of damage to tumors. The mechanism includes (1) increasing the fluidity of tumor cell membrane and causing the destruction of membrane structure and function and (2) destroying the lysosomal membrane and endoplasmic reticulum membrane due to the massive release of lysosomal acid hydrolase, resulting in the rupture of the cell membrane, cytoplasmic spillage, and the death of cancer cells [87]. In Table 2, near-infrared photosensitizers with photothermal properties are collected. The external light source will generate gold nanomaterials’ localized surface plasmon resonance. With the different nanostructures, the absorbance wavelength can be tuned. Gold nanostar [88] possessed absorption peaks at 518 and 680 nm and was combined with caspase-3 imaging in glioblastoma tumors for apoptosis monitoring. Temperature change (∆T) approached 30 °C. The high biocompatible carbon nanodot with multiple functions is another method for cancer imaging and therapy, as shown in Figure 5b. In situ solid-state synthetic methods were used to obtain highly crystalline carbon nanodots with mesoporous carbon nanospheres. The conjugated π electron system was also enhanced to achieve a higher temperature change of around 50 °C with a photothermal conversion efficiency (η) of 43%. This temperature was higher than other nanomaterials (carbon nanospheres, 35%; graphene nanodots, 29%; and gold nanorods, 21%) [89]. However, nanoparticles with magnetic and thermal properties usually have lower thermal conversion efficiency [90,91]. Organic conjugated polymer is another ideal near-infrared absorbing material that can promote effective near-infrared photothermal conversion by reducing its absorption range in NIR-II. The increasing π conjugate length of D-A structured molecules was explored and developed to enhance photothermal conversion efficiency. The 1064 nm laser was compared with the 808 nm laser to distinguish their penetration ability, and 1064 nm was better than 808 nm. Regarding D-A structured organic molecules, the photothermal conversion efficiency exhibited was around 30% with the photoacoustic imaging in glioblastoma treatment [44].

#### 3.2.2. Photodynamic Therapy

Photodynamic therapy refers to the use of light to drive a photosensitizer to produce reactive oxygen species to achieve the effect of local environmental treatment. The drive of photodynamic therapy must have photosensitizer, oxygen, and excitation light sources. When photon energy interacts with oxygen, it will form singlet oxygen, damaging DNA and causing cancer cell death through necrosis or apoptosis [92]. There have been various studies on photosensitizers, such as polymetallic complexes, organic fluorophores, and transition metal complexes in photodynamic therapy [93]. The following works carried out by research groups deal with some of the utilized PS for glioblastoma-based photodynamic study. Zhang et al. worked on ultrasmall DOX-Cu_2−x_Se quantum dots that help generate ROS (namely, ˙OH and ^1^O_2_) using two-electron transfer mechanisms and energy transfer, respectively, as shown in Figure 5c. The most exciting aspect of this fabricated system was the absorption in the NIR-II window, which has higher penetration power in malignant glioblastoma therapy. These studies confirmed that these quantum dots could travel through the blood–brain barrier using ultrasound effectively without any hindrance, helping in photodynamic and chemotherapeutic treatment [83]. Some recent works utilize specific cell-organelle-targeting systems for more efficient photodynamic action. The necessity of this organization is due to the recurrence of tumors post-operation. Organelle targeting becomes a more practical and efficient approach to minimize the reoccurrence of tumors. Vasilev et al. worked on a mitochondria-targeted photosensitizer, namely tetramethylrhodamine methyl ester (TMRM), to evaluate the mitochondria capability [94]. They used a small dose of this photosensitizer approach and observed that this dye was excited by the green/yellow spectrum having better tissue penetration. Moreover, glioblastoma cells were reduced with this dye, as this photosensitizer integrated into the mitochondria of the glioblastoma cells and showed a better photodynamic effect. The organelle-based approach was beneficial for more targeted photodynamic therapy, but it will not be sufficiently effective without drugs. Hence, a drug and photosensitizer associated organization is a more effective method of treating glioblastoma.

#### 3.2.3. Other External Energies-Dependent Therapies

Radiotherapy is also a standard GBM treatment in clinical. Traditional immunotherapy on GBM has many problems, such as immune response tracking, tolerable dose, and side effects. Applying the nanoparticle for tracking agents or vehicles for delivery may be another solution to the challenges. Radiotherapy is the most frequent treatment option for GBM patients in clinics. However, the side effects are unimaginable. It cannot be effective treatments of GBM that prefer to combine other treatments or nanoparticles that can bring a better synergetic result and lower side effects on GBM patients [20]. Cold atmospheric plasma is another new technique applied to cancer therapy. A non-thermal plasma (ionized gas) can trigger nanoparticle uptake and accumulate in the targeted tumor site. In the research, cold atmospheric plasma has been applied to the GBM therapy with the nanoparticle to obtain better synergistic cytotoxicity against GBM cells.

## 4. Nanocomposites Combined with Chemotherapy Applied in Curing Glioblastoma

Nanoparticle-based drug delivery has long been considered a better and more reliant transfer drug. The vessel vasculature of normal and tumor tissue varies significantly, where tumor vessels are more exhaustive and open than usual [101]. This difference in the diameter of the vessels contributes to the application of targeted drug delivery. Moreover, the arteries, capillaries, and veins are not connected, as normal tissues observe. This complex architecture makes the behavior of tumor tissues more unpredictable, for which only specific mechanisms of drug delivery are acceptable. In some cases, the embedded drugs flow through a nanoparticle of around 100 nm, which is heavier and cannot diffuse properly to the tumor area. However, smaller nanoparticles of 1–10 nm can directly help the drugs in being targeted to the tumor area. Gliomas account for a significant number of deaths worldwide. Some 30% of the primary brain tumors responsible for the most brain-related deaths are gliomas. Histologically classified gliomas include oligodendrogliomas, astrocytomas, ependymomas, conjugating oligodendrogliomas, astrocytomas [102], and WHO grade IV gliomas, which affect elderly patients above 50 years and may occur in children and young adults too. There has been a rise in the number of FDA-approved anticancer drugs in the past 20 years. However, the main problem with these oncology drugs is that their development should specifically act on the molecular pathways responsible for tumor metastases and their growth [102].

Moreover, they are poorly soluble in aqueous solutions and have a small therapeutic window that restricts their ability effectively to treat cancer alone. Thus, the role of nanoparticle-based drug delivery seems viable for targeted tumor therapy. Treating glioma requires the drug to cross the blood–brain barrier (BBB) and reach the tumor site. However, it is tough to pass through the BBB as it is dominated by pericytes, endothelial cells, and astrocytes that obstruct smooth drug delivery. Although BBB changes to have improved permeability with tumors, it is not enough for the optimum transfer of drugs inside the tumor region [103]. This section will summarize how chemotherapeutic drugs and nanomaterials work together to treat GBM. It also discusses what kinds of nanomaterials can be used as candidates after surface modification and what loading drugs can be applied for GBM chemotherapy. We propose some examples to validate this research field at the end of this section (Figure 6).

### 4.1. Composition of Nanoparticles

#### 4.1.1. Liposomes

Liposomes, with high biocompatibility and capability of encapsulating both hydrophobic and hydrophilic substances, have become the most commonly used chemotherapeutic nanocarriers for GBM [104]. Lipids with PEG chains are often taken as membrane components to facilitate penetration through BBB and the follow-up modification [105]. In a 2020 study, the authors provided evidence that nanoliposome-encapsulated oleanolic acid has emerged as having a robust proliferation effect on U87 cells, 40% that of free oleanolic acid at a concentration of 35 μmol/L [106]. As a marker of the invasive ability of cancer cells, vimentin exhibited an improper distribution and a significant decrease when treated with nanoliposome-encapsulated oleanolic acid, indicating the formulation has effectively reinforced the suppression effect of migration [106,107].

#### 4.1.2. Solid Lipid Nanoparticles

Solid lipid nanoparticles are spherical nanocarriers with solid lipid cores, which exhibit a high affinity to hydrophobic medicine [108]. In addition, solid cores can mitigate the chronic toxicity of drugs. Solid lipid nanoparticles can be classified into neutral, cationic, and anionic. In contrast, cationic liposomes have drawn the most excellent attention due to significant penetration through BBB by adsorption-mediated-transcytosis [109]. Banerjee et al. synthesized PEG solid lipid nanoparticles (PEG-SLN). They found that the antiangiogenic capability and PTX-based glioma cells apoptosis were enhanced, which was impossible if the drug was used alone. 

#### 4.1.3. Cubosomes

Cubosomes are cubic liquid crystalline nanoparticles with sizes ranging from 10 to 500 nm, formed by the self-assembly of surfactant-like lipids [110]. The complexity of this nanostructure endowed itself with a large surface area, enabling cubosomes to achieve higher loading capacity and desirable drug release performance than simply round nanoparticles [111]. In a 2020 study, the drug entrapment efficiency and drug loading capacity of cubosomes were 97.7% and 9.9%, which are relatively high compared to other previously reported nanocarriers [112]. The treatment of cubosome-incorporated AT101 has improved cytotoxicity of LN229 by 20% than treated by free AT101, manifesting the promise of cubosome in the treatment of GBM [113].

#### 4.1.4. Polymeric Nanoparticles

Polymeric nanoparticles utilized in drug delivery to the brain are often manufactured from biodegradable polymers, such as poly(lactic-co-glycolic acid) (PLGA), poly(lactic acid) (PLA), and poly(glycolic acid) (PGA), for instance [114]. The degradation products of these polymers are lactic acid and glycolic acid, which turn into carbon dioxide and water after the Krebs cycle [115]. Polymeric nanoparticles possess a better release profile and there is no need to be concerned over oxidation issues as lipids are much more stable [116]. Nance et al. formulated a polyethylene glycol (PEG) coated PLGA nanoparticle complex. They found that 70 nm nanoparticles passed quickly inside the tumor parenchyma, resulting in more significant tumor growth suppression [117].

#### 4.1.5. Silica Nanoparticles

Synthesized mesoporous silica will be used as a substrate, and the synthesis procedure for the nanosystem is displayed in Figure 7a. The Cr^3+^- and Sn^4+^-codoped zinc gallate in mesoporous silica nanoparticles (MSNs) were first developed and used as persistent luminescent nanoparticle (PLN) nanocarriers. The original MSN is approximately 50 nm and has a distinct pore structure. After sintering at high temperatures, the holes grow into a dark-colored ZnGa_2_O_4_:Cr^3+^, Sn^4+^ core. Calcination was performed at 1000 °C to make a PLN sphere. NB was formed using three kinds of lipids. The nanobubble (NB) precursor solutions of DPPC, DSPE-PEG2000, and DPPA mixed with molecular drugs such as TMZ were generated. PLN was linked by electrostatic force and first functionalized with an amino group to achieve a positive charge on the surface. Aptamer AS1411 (AAp) was covalently conjugated with PLN to target the GBM cells. This conjugation was obtained via the amide coupling of the carboxyl group of the aptamer and the amino group of TMZ–NB@PLN. After hybridization with NB and the targeting ligand, TMZ-NB@PLN-AAp was obtained. This nanosystem can accumulate in GBMs through AAp targeting. Ultrasound induction opened BBB for a brief time. TMZ was released from the NB and passed through the BBB to treat GBMs. The performance of TMZ–NB@PLN–AAp in mouse brain was monitored and observed. Three different groups, namely, control, TMZ–NB@PLN, and TMZ–NB@PLN–AAp, were injected with the drug to treat GBM cells. After the 3-week treatment, GBM shrank in size for TMZ–NB@PLN and TMZ–NB@PLN–AAp groups, as shown in Figure 7b,c [75].

### 4.2. Surface Modification

Even though most of the nanocarriers used in therapy for GBM are made of a phospholipid, which is expected to pass through BBB, the penetration rate is relatively low. Moreover, most nanoparticles can only accumulate passively in tumors via enhanced permeability and retention (EPR) effect. Accordingly, nanovehicles should be modified with functional ligands to acquire a higher penetration rate through BBB and better tumor-targeting ability. Some ligands correspond to which receptors are overexpressed on BBB. Others are associated with receptors overexpressed on brain glioma cells; still, others can promote targeting or possess particular traits. 

#### 4.2.1. Folate Receptor (FR) Targeted Ligand

Folate, one of the B vitamins, plays a crucial role in DNA and RNA synthesis and is responsible for the proliferation of cells. Thus, the uptake of folate is vital in that folate receptors are often highly expressed on the surface of brain glioma cells. As a strategy to cause drug accumulation in tumor cells, FR-targeted ligand is usually applied to the surface of nanoparticles. Kim et al. worked on an immunoliposome nanocomplex with the envelopment of MPEG-PLA. Folate acid has noticeably crippled the activity of cancer cells because of good binding activity to folate receptors attached to the cell membrane of tumor cells [118]. 

#### 4.2.2. Transferrin Receptor (TfR) Targeted Ligand

Since transferrin is associated with the activation of cell division, it is a vital protein for cancer cells. Transferrin receptors (TfR) are overexpressed on BBB and brain tumor cells, making Tf a common ligand for the GBM therapeutic drug system. However, there are better TfR targeted ligands than Tf for the high concentration of endogenous Tf might prevent the binding of Tf modified on nanocomposites with TfR. Ligands with different binding sites from Tf, such as HAIYPRH (T7), are deemed as potential candidates. 

#### 4.2.3. Glucose Transporter Targeted Ligand

Glucose transporters are highly expressed on endothelial cells of brains [119,120]. P-amino-phenyl-α-D-manno-pyranoside (MAN) was manifested to assist nanoliposomes in passing through the BBB successfully due to its high affinity to glucose transporters 1 (GLUT1), which is highly expressed on BBB and glioma cells [121]. In addition, the charges it carries have increased the electrostatic repulsion between particles, making them more stable in solution.

### 4.3. Nanoformulations with FDA-Approved Drugs

#### 4.3.1. Paclitaxel

Paclitaxel (PTX) is an excellent anticancer agent that stops cell growth and further division by hindering microtubules from assembling in the cell [122]. In recent years, the co-loading of dual drugs has been studied. Madani and Maleki et al. worked on dual-cure system-based delivery with different polymer NPs [123,124]. Wang et al. worked on a polycaprolactone (PCL)-PEG-PCL system with PTX and irinotecan (SN38) as represented in Figure 8a and showed by in vivo and in vitro studies that PTX@SN38-integrated nanomaterials were better at reducing tumor formation for a more extended period as shown in Figure 9a–d [125]. This formulation is more effective as there is a better chemotherapeutic effect and less drug resistance. Additionally, two drugs reduce the risk of a decrease in drug transfer to the tumor region [126,127]. 

#### 4.3.2. Doxorubicin

Doxorubicin (DOX) hinders the growth of cancer cells by damaging the topoisomerase-II-mediated DNA repair after integration into DNA. [128]. Dhar et al. worked on the system using sophorolipid, a glycolipid verified as an anticancer agent [129,130]. They created the nanoparticle system using gellan-gum and gold nanoparticles, identified as non-cytotoxic and loaded with DOX for killing glioma cell lines [131]. Zhong et al. treated a PEG-PCL composite polymer with cRGD-Au nanorods and found more outstanding targeted drug release. A NIR laser helped activate the release of the drug into the site of action with better effectiveness [132]. A similar mechanism was followed by Ruan et al. for receptor-mediated drug delivery [133]. In recent years, the dual-targeting method of the ligand attachment and high-intensity focused ultrasound (HIFU) has been studied by Luo et al. where HIFU-responsive angi-opep-2-integrated composite PLGA NP drug delivery material holding DOX/ PFOB (ANP-D/P) was synthesized, as shown in Figure 8b, for improving the overall drug acceptance, release, and the targeting of glioma cells. The in vivo study conducted in Figure 9e shows H&E staining images where improved pyknosis and karyorrhexis for ANP-D/P hybrid were observed rather than saline or other groups. Additionally, in Figure 9f, effective apoptosis was observed for angiopep-modified systems with drugs [134].

#### 4.3.3. Temozolomide 

Temozolomide (TMZ) acts as a DNA alkylating agent by affecting the cell cycle, promoting cell arrest in the G2/M phase that causes cell apoptosis [135]. Dilnawaz et al. worked on the superparamagnetic nanoparticles system loaded with TMZ and curcumin for the dual therapy of glioma, using the glioma spheroid model and 2D monolayer system showed that this system enabled an anticancer effect in both models [136]. 

### 4.4. Nanoformulations with Natural Compounds

In addition to thoroughly-studied FDA-approved medicines, scientists consider many plants’ secondary metabolites potential candidates for curing glioma. Plant’s secondary metabolites (PSMs), which are defined as compounds that affect plant growth indirectly, often possess pharmaceutical activities owing to their distinctive skeletons and large functional groups. However, they have not been medically utilized due to their unclear anticancer mechanism. Moreover, their low water solubility, inefficient cancer cell-targeting ability, and poor bioavailability become hurdles to their practical use. As a result, the research related to PSMs is often focused on testing PSMs’ effect on tumor-related RNA and protein. In addition, the topics are usually conjugated with nanodrug delivery systems or synergetic development with chemotherapy drugs, promoting their efficacy in the treatment of glioma. Compared to FDA-approved medicines, PSMs’ usage and biological pathway are often attached with more importance than the formulation of nanoparticles in the findings related to PSMs combining nanodrug delivery. According to their structures, PSMs can be roughly classified into terpenes, polyphenolics, and alkaloids.

#### 4.4.1. Terpenes and Terpenoids

Terpenes and terpenoids are the most prominent families among PSMs, providing plants with a great range of physiological functions [137]. Terpenes are organic compounds solely composed of isoprene, which is a five-carbon building block, while terpenoids are derivatives of terpenes [138]. A wide variety of terpenes and terpenoids have been demonstrated to cure glioma [139,140,141]. For instance, oleanolic acid, a pentacyclic triterpenoid acid naturally occurring in many plants, was revealed to exhibit antitumor effects by suppressing proliferation, migration, and invasion abilities [139,140]. Ursolic acid, a pentacyclic triterpenoid abundant in herbs, has recently drawn researchers’ attention due to its ability to activate multiple signaling pathways [141,142,143,144]. Ying et al. coloaded ursolic acid and epigallocatechin 3-gallate (EGCG) as anticancer agents into liposomes. The synergistic effect of ursolic acid and EGCG as individual free drugs was manifested by the inhibition rate, which can be attributed to ursolic acid arresting the cell cycle at the G2 phase while EGCG is arresting the cell cycle at the G0/G1 phase, restraining the mitosis and synthesis of RNA and protein [145,146]. 

#### 4.4.2. Polyphenols

Polyphenols are natural compounds with several hydroxyl or carbonyl groups on their aromatic rings [147]. They are often responsible for the color of the plants, which is attributed to the abundant bonds in their structures [148]. Since these kinds of compounds are mostly produced to defend against unfavored environments for plants, their biochemical abilities have provided a complete forecast for the development of anticancer drugs [149,150]. Polyphenols have been reported to exert anti-tumor effects via various pathways, such as suppressing cancer cell proliferation and migration, enhancing reactive oxygen species (ROS), scavenging enzyme activities, and inducing apoptosis [151,152]. Resveratrol is a rising star in anti-cancer agents because it can inhibit tumor cells via a broad spectrum of pathways [153,154,155]. In a study conducted by Xu et al., TMA/resveratrol coloaded mPEG-PCL nanoparticles were applied to U87 cells to demonstrate the synergistic effect of the two medicines [156]. The downregulation of the antiapoptotic protein Bcl-2 and the upregulation of the proapoptotic protein Bax were characterized by the inhibition of p-Akt expression. 

#### 4.4.3. Alkaloids

Alkaloids are natural compounds containing nitrogen atoms in the aromatic rings, often derivatives from amino acids [157]. Even though most alkaloids are neurotoxins to human beings, some of them have already been approved by the FDA, such as vincristine, atropine and camptothecin derivatives [158,159,160,161]. Trigonelline, a significant constituent of *Trigonella foenum-graecum,* is reported to be apoptotic and anti-invasive via the downregulation of nuclear factor E2-related factor 2 (Nrf2) dependent proteins and proteasome-dependent proteins [162]. 

### 4.5. Drugs and Nanoparticles Complex Platform

After conjugation of drugs into nanoparticles, the complexity and dynamics associated with the involvement of the medicines and nanoparticles are significant aspects that erquire evaluation. Several bonds are responsible for adhering drugs to nanoparticles. Li et al. worked on camptothecin-DOX-loaded MSN, wherein DOX was joined using acid-labile hydrazone bonds, and camptothecin was filled inside the MSN pores [163]. There are also multiple bonds involved in maintaining the drug and nanoparticle system; Ruan et al. focused on gold nanoparticle-based drug delivery-system where DOX was modified by potassium thioacetate and acrolein to form S-3-oxopropyl ethanethioate. Then, hydrazine was produced using aldehyde and carboxyl group reaction. DOX interaction further helped in the formation of the hydrazone linker. Thiol group was formed after deacetylation. Finally, this thiol group was attached with the gold nanoparticle, using thiol–gold interaction, as shown in Figure 10a [164]. The biocompatibility, biodegradable nature of iron oxide nanoparticles (IONPs), and their flexibility to be modified with several conjugation systems have been studied by various researchers for delivering the drug to the brain. Norouzi et al. utilized Trimethoxysilylpropyl-ethylenediamine tri acetic acid (EDT) to stabilize the IONPs, due to its property of imparting negative charge. This negative charge then interacts with DOX using ionic bonding. Moreover, cadherin binding peptide ADTC5 on the DOX-EDT IONPs helped increase permeability through BBB in the presence of an external magnetic field. Lipid nanoparticles are one of the most versatile materials to be reproduced and modified, as observed in the study put forward by Banerjee et al., where he altered the solid lipid nanoparticles (SLN) with Tyr-3-octreotide (TOC). PEG was conjugated with TOC using a resin-based reaction in dimethylformamide (DMF). Then, the resins were reacted using a coupling reaction with 1,2-dipalmitoyl-sn-glycerol-3-phosphoethanolamine (DPPE), the lipid. The reaction proceeded with N-hydroxysuccinimide (NHS) and 1-(3-dimethyl aminopropyl)-3-ethyl-carbodiimide (EDC) reacting in DMF for 4h activating the acid group. Then, the activated resins were mixed with DPPE in the chloroform and methanol mixture (9:1), respectively. Then, PTX was dissolved in poloxamer solution, and then this solution was mixed with the lipid nanoparticles for surface modification. This system had TOC, which had an affinity towards somatostatin receptors (SSTR2) and eventually with drug helped in anti-glioma therapy. The targeted pathway system relies on specific antibodies responsible for attaching to the receptors concerning glioma therapy. A study shown by Banstola et al. showed the conjugation of panitumumab (PmAb) antibody with TMZ functionalized on PLGA Nps. The chemistry involved is EDC and NHS coupling chemistry, as shown in Figure 10b. They activated the carboxyl end of PLGA-NPs, stabilized using continuous stirring for 20 min, and then PmAb was mixed with activated NPs. This construct successfully helped in more apoptotic cell death by starting caspase-9 and downregulation of LC3B and Beclin-1 [165]. Herein, we research the varied formulations of drugs with nanoparticles, as shown in Table 3.

## 5. Discussion and Conclusions 

The primary purpose of nanomedicine is divided into two parts: diagnosis and treatment. Nanomedicine has extraordinary capabilities and effects, but it also has related problems. First, carefully designed nanomaterials have unique physical and chemical properties that can diagnose the location of GBM and provide imaging information. Whether through MRI or NIR, nano-particles can show their characteristics well and are used in current medical diagnoses. However, the degradability of nanomaterials is a significant test for a developer. When many nanoparticles accumulate in body organs, it may cause harm and damage. The second is to package the drug into nanoparticles and deliver it to a specific target in the human body. When a nanoparticle is used as a delivery medium, the type of drug loaded onto it is essential because differently loaded drugs have extra total load and drug release kinetics. However, brain tissue is a very challenging environment for drug administration. Part of this is the need to prevent harmful substances from entering the brain through the blood. The brain has the so-called BBB mechanism, which means that only certain substances are selectively allowed to enter the brain. In addition, the space of the brain cells is quite cramped and small. This review summarizes the insights that various studies have brought to nanomedicine. It is hoped that research in this field can further improve existing problems and enter the clinical stage as soon as possible to enhance human wellbeing.

## Figures and Tables

**Figure 1 pharmaceutics-14-00456-f001:**
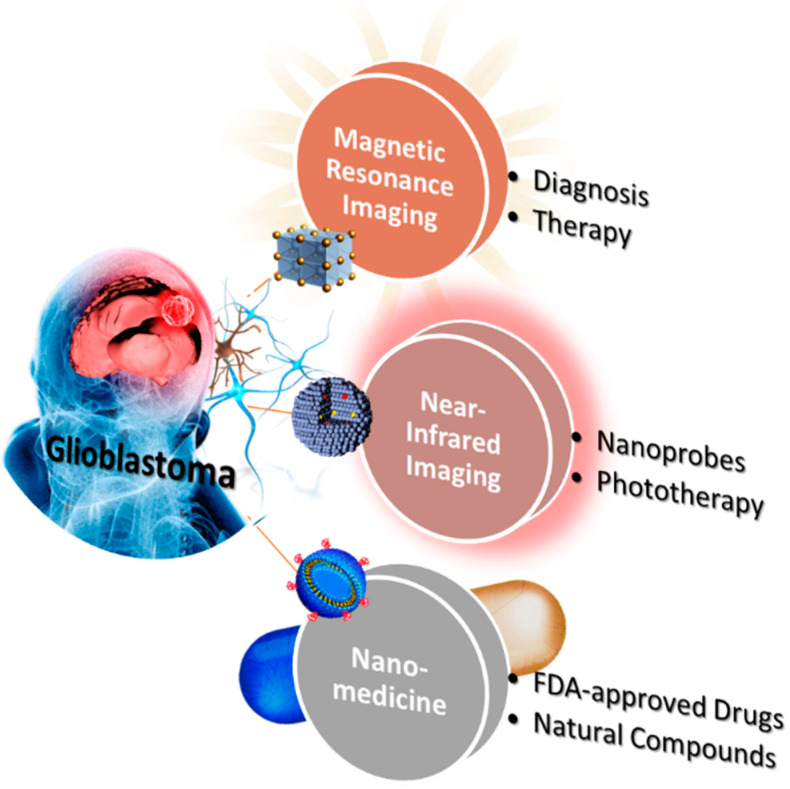
The three primary goals of research into GBM. The current research focuses on the long-term magnetic and near-infrared diagnosis and natural and chemical drugs equipped with nanomaterials.

**Figure 2 pharmaceutics-14-00456-f002:**
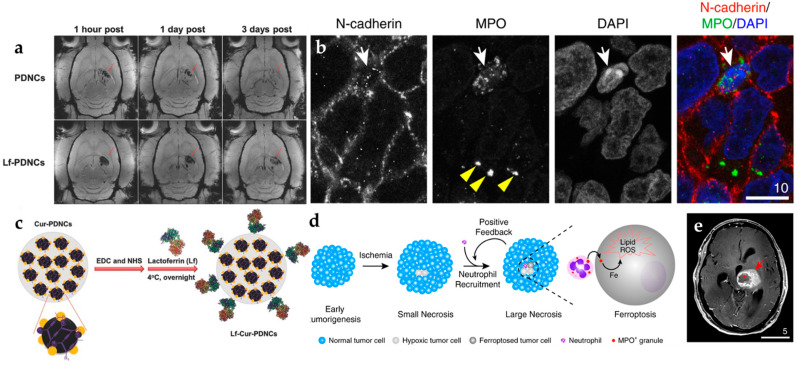
The iron-based nanoprobes in glioblastoma MRI imaging. (**a**) T2-weighted MRI images with contrast accumulation in GBM [25]. (**b**) The confocal images demonstrate that the accumulation of MPO was located in the cytosol (with FITC tracking dye) [26]. (**c**) Modified with lactoferrin on the surface of iron nanoparticles. (**d**) Illustration of neutrophil recruitment crossing with targeted magnetic iron nanoparticles (MPO) and (**e**) in brain tissue GBM MRI imaging. Adapted with permission from Refs. [25,26], Copyright 2016 Wiley and 2020 Springer.

**Figure 3 pharmaceutics-14-00456-f003:**
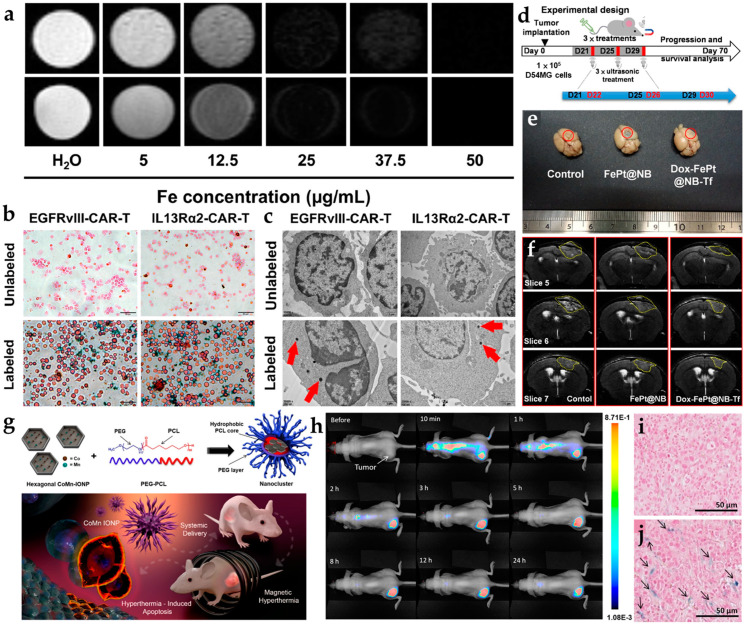
The various types of iron-based nanoplatforms in glioblastoma theranostics. (**a**) T2-weighted images containing iron-oxide nanoparticles labeled CAR-T cells, EGFRvIII CAR-T (upper row), and IL13Rα2 CAR-T (lower row) cells. (**b**) There are representative images of Prussian-blue-stained, unlabeled CAR-T cells (top) and labeled CAR-T cells (bottom). (**c**) Cell TEM views of unlabeled CAR-T cells (top) and labeled CAR-T cells (bottom) [18]. (**d**) Illustrative mice model of synthesis and HIFU-triggered drug release from FePt@NB. (**e**) Photos of mouse brains with GBMs. The GBM tumors are circled with a yellow highlight. (**f**) T2-weighted MRI images of mouse brains with GBMs. The GBM tumors are highlighted in yellow circles [32]. (**g**) Graphical representation of CoMn IONP encapsulated inside PEG/PCL polymers. (**h**) IVIS system analyzed for injection of CoMn-IONP nanoclusters loaded with a hydrophobic NIR dye. Prussian-blue staining of tumor slices of (**i**) 5% dextrose and (**j**) CoMn-IONP nanoclusters [34]. Adapted with permission from Refs. [18,32,34], Copyright 2021 Elsevier, 2019 and 2021 American Chemical Society.

**Figure 4 pharmaceutics-14-00456-f004:**
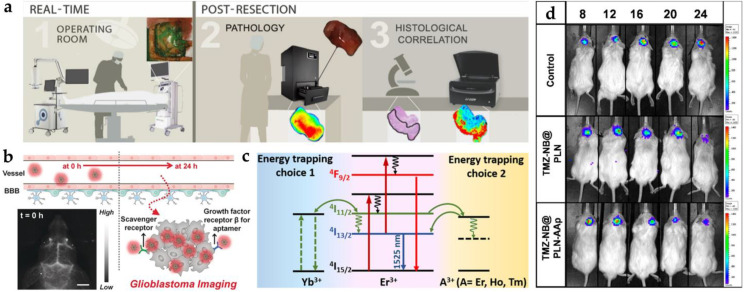
The common nanoprobes in glioblastoma imaging. (**a**) Three kinds of workflow in glioblastoma imaging with cetuximab-IRDye800D [48]. (**b**) Illustration of blood–brain barrier crossing with targeted NIR-II fluorescence in brain vessel imaging [64]. (**c**) Mechanism of energy transfer with Yb^3+^, Er^3+^, and A^3+^ downconversion system [69]. (**d**) The IVIS image from persistent luminescent ZnGa_2_O_4_:Cr^3+^, Sn^4+^ with the tumor change (days 8, 12, 16, 20, and 24) [75]. Adapted with permission from Refs. [48,64,69,75]. Copyright 2018 Springer, 2020 Wiley, 2020 Elsevier and 2021 American Chemical Society.

**Figure 5 pharmaceutics-14-00456-f005:**
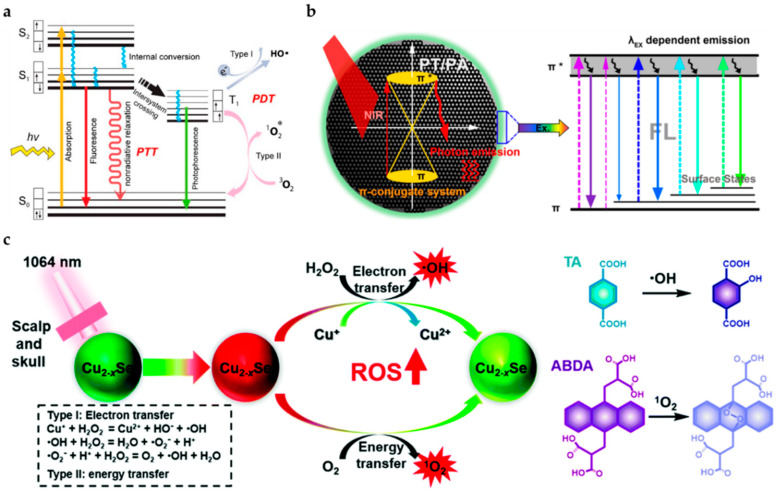
The common nanoprobes in glioblastoma imaging. (**a**) The common pathways of photothermal and photodynamic therapy [85]. The * symbols next to the atomic or atomic group indicates that there are no paired electrons. (**b**) The mechanism of carbon nanodot with simultaneous laser irradiation to induce photothermal effect and fluorescence [89]. (**c**) The two ROS generated ultrasmall Cu_2−x_Se quantum dots by electron transfer and energy transfer [83]. Adapted with permission from Refs. [83,85,89]. Copyright 2009 Royal Society of Chemistry, 2018 and 2020 American Chemical Society.

**Figure 6 pharmaceutics-14-00456-f006:**
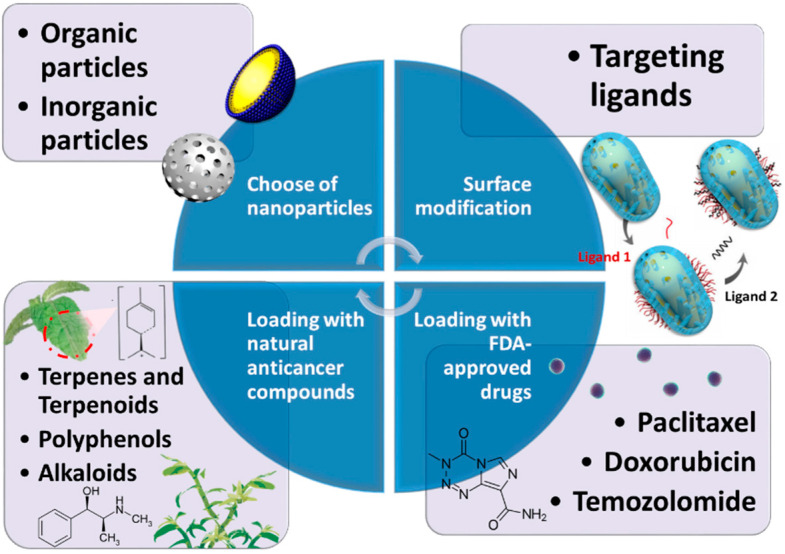
Schematic illustration of the nanocomposites combined with chemotherapy. The drug delivery system for GBM chemotherapy can be optimized by the composition of nanoparticles, ligands, and medicine.

**Figure 7 pharmaceutics-14-00456-f007:**
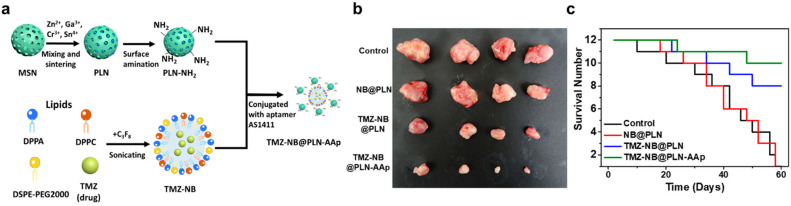
(**a**) Schematic illustration of nanosystem preparation and surface functionalization [75]. (**b**) The tumor sizes after different treatments. (**c**) The change in the survival number of the mouse over time. Adapted with permission from Ref. [75]. Copyright 2021 American Chemical Society.

**Figure 8 pharmaceutics-14-00456-f008:**
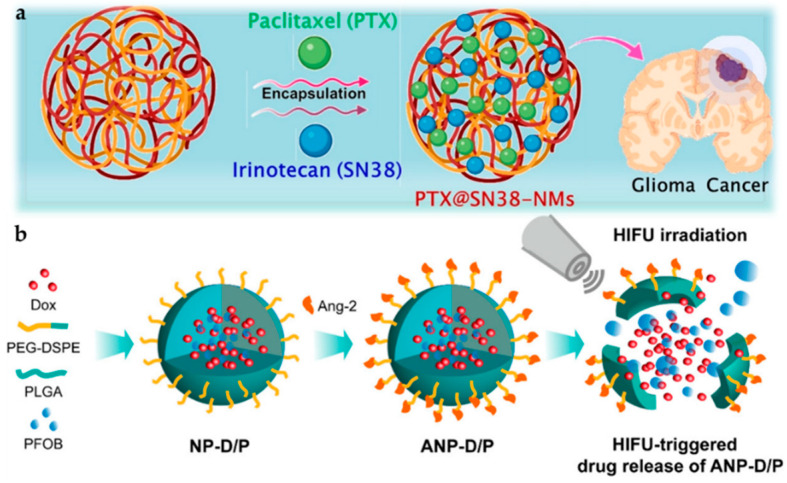
Scheme of various nanoparticle systems with embedded drugs. (**a**) Graphical representation of PTX and SN38 encapsulated inside PCECs [125]. (**b**) Illustrative model of synthesis and HIFU-triggered drug release from ANP-D/P [134]. Adapted with permission from Refs. [125,134]. Copyright 2021 Elsevier and 2017 American Chemical Society.

**Figure 9 pharmaceutics-14-00456-f009:**
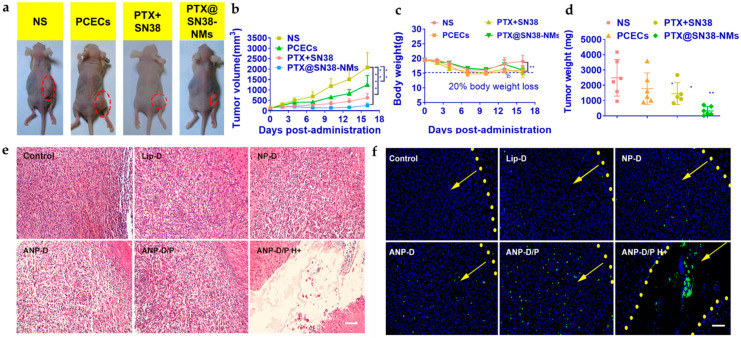
In vivo antitumor efficacy of nanoparticle-drug system (**a**) shown for different systems of NS (control), PCECs (composite), PTX@SN38 (dual-drug), and PTX@SN38-NMs. (**b**) Tumor volume (mm^3^), (**c**) body weight loss study, and (**d**) tumor weight (mg). The data are represented as the mean ± SD; * *p* < 0.05, ** *p* < 0.01, and *** *p* < 0.001 [125]. Anti-glioblastoma efficacy shown by (**e**) H&E staining and (**f**) TUNEL assay of U87 MG glioblastoma tissues treated with different Dox formulations (Lip-D, NP-D, ANP-D, ANP-D/P, and ANP-D/P H+). Green represents the TUNEL-positive cells; blue, cell nuclei; and the yellow dashed line, the border of the glioblastoma. The yellow arrows indicate the direction of the glioblastoma (scalebar = 100 μm) [134]. Adapted with permission from Refs. [125,134]. Copyright 2021 Elsevier and 2017 American Chemical Society.

**Figure 10 pharmaceutics-14-00456-f010:**
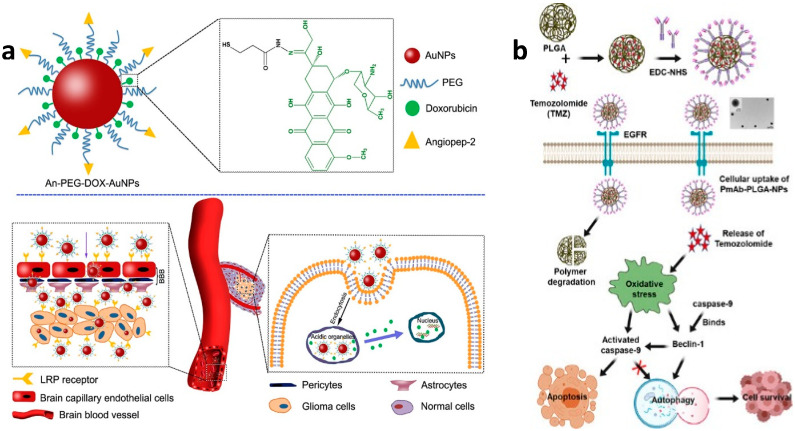
Interpretation of the drug-nanoparticle complex formation in (**a**) An-PEG-DOX AuNPs [133] and (**b**) PmAb-TMZ-PLGA-NPs [181]. Adapted with permission from Refs. [133,181]. Copyright 2015 Elsevier and 2020 American Chemical Society.

**Table 1 pharmaceutics-14-00456-t001:** The in vivo NIR imaging with different nanoprobes.

Type	NIR Windows	Nanoprobe	Dose	Emission	Ref.
Organic dye	NIR-I	IR dyes	1 μM *	650–950 nm	[42,43,44,45,46,47,48,49,50,51,52]
		Cy5.5	2 mg/mL	690–750 nm	[53,54,55,56,57]
		ICG	200 nmol/kg	750–950 nm	[58,59]
		MMP	20 nM	750 nm	[60]
		SiNC	1 mg/kg	600–800 nm	[61]
	NIR-II	New designed IR dyes	10 mM	900–1400 nm	[62,63,64]
Lanthanide-doped nanoparticle	NIR-I	Tm-doped	15 mg/kg	800 nm	[65]
	NIR-II	Nd-doped	1 mg/mL	1060 and 1340 nm	[66,67,68]
	NIR-III	Er-doped	5 mg/mL	1525 nm	[69]
Quantum dot	NIR-I	Ag_2_S	1 mg/mL *	650–840 nm	[70]
	NIR-II			1000–1400 nm	[71,72,73]
		N,B-doped graphene quantum dot	1 mg/mL	950–1100 nm	[74]
Nanophosphor	NIR-I	Cr^3+^-doped	250 mg/mL *	650–850 nm	[75]
Polymer dot	NIR-II	Aggregation-induced emission (AIE)	10 mg/kg *	800–1600 nm	[76,77,78,79]

***** The dose concentration is selected from one of the references.

**Table 2 pharmaceutics-14-00456-t002:** The in vivo phototherapy with different photosensitizers.

Therapy	Type	Photosensitizer	Excitation Wavelength	Dose	Approach	Ref.
Photothermal	gold nanomaterial	gold nanostar	808 nm laser (2 W/cm^2^)	10 nM	ΔT ^1^ ~ 30 °C	[88]
	two-dimensional materials	mesoporous silica-coated graphene nanosheet	808 nm laser (6 W/cm^2^)	50 μg/mL	ΔT ~ 30 °C	[95]
	quantum dot	carbon nanodot	808 nm laser (2 W/cm^2^)	5 mg/mL	ΔT ~ 50 °C; η ^2^ = 42%	[89]
		N,B-doped graphene quantum dot	808 nm laser (1.5 W/cm^2^)	1 mg/mL	η = 32%	[74]
	metal oxide materials	Fe_3_O_4_@Au	635 nm laser (0.3 W/cm^2^)	0.5 mg/mL	ΔT ~ 20 °C	[90]
		Mn-doped magnetic nanoclusters	750 nm laser	250 μg/mL	ΔT ~ 16 °C	[91]
	organic dye	COF@IR783	808 nm laser (0.75 W/cm^2^)	2.5 mg/kg	ΔT ~ 20 °C	[96]
		D–A structured molecular	1064/808 nm (1 W/cm^2^)	0.05 mg/kg	η = 30%	[77]
		ICG	808 nm (2 W/cm^2^)	40 μg/mL	η = 45%	[97]
		ApoE-Ph (AIE)	808 nm laser (0.5 W/cm^2^)	10 mg/kg	η = 34%	[79]
Photodynamic	quantum dots	Cu_2−x_Se	1064 nm laser (0.75 W/cm^2^)	5 mg/kg	n/a	[83]
	organic dye	dicysteamine-modified hypocrellin derivative (DCHB)	721 nm laser (0.5 W/cm^2^)	0.5 mM	η = 33%; quantum yield of 0.51	[98]
	inorganic nanoparticle and organic dye	ANG-IMNPs	PTT: 808 nm laser (0.36 W/cm^2^) PDT: 980 nm laser (0.8 W/cm^2^)	1.1 mg/kg	ΔT ~ 18 °C; cell viability ~ 28%	[99]
		Fe_3_O_4_-IR806	808 nm laser (3 W/cm^2^)	10 mg/kg	η = 42%; cell viability ~ 19.2%	[100]

^1^ ΔT is the temperature in change. ^2^ η is photothermal conversion efficiency.

**Table 3 pharmaceutics-14-00456-t003:** Chemotherapy applied in glioblastoma treatments with different nanoparticles.

Nanoparticles (NP)		Medicine	Diagnostic Methods	Dose	Targeted Cellular Pathway	Ref.
Lipid NP	Dioleoyl phosphatidylcholine (DOPC), Cholesterol (Chol), 16-DCL, PD-L1siRNA, and DSPE-PEG(2000)amine	PTX	Liquid chromatography-mass spectrometry (LC-MS)	20 μM	n/a	[166]
Cubosomes	Glyceryl monooleate and Pluronic F-127	AT101	UV/Vis/NIR spectrophotometry and qRT-PCR	10 wt%	Akt-signaling pathway	[167]
Polymer NP	PLGA	Camptothecin	H&E staining and IHC staining	180 mg/kg	DNA damaging	[168,169]
Catanionic solid lipid NP	Anti-epithelial growth factor receptor (EGFR)	Carmustine	HPLC-UV system and enzyme-linked immunosorbent assay (ELISA)	1.5 mM	PI3K/ AKT signaling pathway	[109,170]
Brain penetrating NP	Polyaspartic acid (PAA) and PEG	Cisplatin	MRI and ultrasound	40 wt%	TNFα pathway	[171,172]
Polymer	PLGA and PEG	Docetaxel	ELISA, LC-MS, and flow cytometry	1500 μg/mL	Arrestment of G2 and M phase	[173,174]
Liposomes	Transferrin and penetratin	Doxorubicin	HPLC, H&E staining	15.2 μmoles/kg	n/a	[175]
		Erlotinib	HPLC and H&E staining		n/a	[175,176]
Hybrid NP	Angiopep-2, PLGA, and perfluorooctyl bromide (PFOB)	Doxorubicin	High intensity focused ultra-sound (HIFU), fluorescence imaging, and flow cytometry	50 μg/mL	n/a	[134]
Magnetic silica NP	Transferrin, PLGA, and mesoporous silica nanoparticle (MSN)	Doxorubicin	Flow cytometry	400 μg/mL	n/a	[177]
		Paclitaxel	Non-invasive bioluminescence imaging and H&E staining			[177]
Lanthanum oxide (La_2_O_3_) NPs	La_2_O_3_	Temozolomide	Western blotting	100 µg/mL	Arrestment of G2/M phase	[178]
Lipid NP	Cholesterol	oleanolic	RT-PCR	17.5 mM/kg	Caspase-3 pathway	[106]
Cubosomes	DiI	UA EGCG	LC-MS	5 mg/kg	Hindrance of mitotic spindle	[179]
Polymer NP	NH_2_-PEG_2000_-DSPE	TMZ resveratrol	CLSM	500 μg/mL	Inhibition of the p-Akt expression	[156]
Catanionic solid lipid NP	mPEG-PCL	Curcumin	flow cytometry	50 mg/kg	Suppression of neovascularization	[118]
Brain penetrating NP	mPEG-PLA	Trigonelline	fluorescence microscope flow cytometry analysis	34 μg/mL	Downregulation of the Nrf2	[180]

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
