# Peer review of "Progress and Viewpoints of Multifunctional Composite Nanomaterials for Glioblastoma Theranostics"

_pharmaceutics, 2022, doi:10.3390/pharmaceutics14020456_

Round 1

Reviewer 1 Report

In the present manuscript “Progress and viewpoints of multifunctional composite nanomaterials for glioblastoma theranostics”, the authors discuss the application of various nanocomposites in diagnosing and treating GBM. However, in my opinion, that this manuscript in its present form is not yet ready for publication. For the favor of improving the paper, some points are listed below:

1) I have found a similar review (Pourgholi F, Hajivalili M, Farhad JN, Kafil HS, Yousefi M. Nanoparticles: Novel vehicles in treatment of Glioblastoma. Biomed Pharmacother. 2016 Feb;77:98-107. doi: 10.1016/j.biopha.2015.12.014. Epub 2015 Dec 29. PMID: 26796272)

So, the authors need to outline a detailed examination of the challenges of GBM treatment and future prospectus of this review

2) In tables, please add the cellular pathways that are targetted through the treatment together with the references

3) Authors should include other types of recent combinational therapy with nano-particles such as radiotherapy (Pourgholi F, Hajivalili M, Farhad JN, Kafil HS, Yousefi M. Nanoparticles: Novel vehicles in treatment of Glioblastoma. Biomed Pharmacother. 2016 Feb;77:98-107. doi: 10.1016/j.biopha.2015.12.014. Epub 2015 Dec 29. PMID: 26796272); cold atmospheric plasma (He, Zhonglei et al. “Cold Atmospheric Plasma Stimulates Clathrin-Dependent Endocytosis to Repair Oxidised Membrane and Enhance Uptake of Nanomaterial in Glioblastoma Multiforme Cells.” Scientific reports vol. 10,1 6985. 24 Apr. 2020, doi:10.1038/s41598-020-63732-y); etc

4) I think, it will be interesting to mention about the dose (in concentration) of nanoparticles that are used in the studies in tables

Author Response

Response to reviewer 1:

No

Reviewer 1 comments

Changes to the manuscript

1

I have found a similar review (Pourgholi F, Hajivalili M, Farhad JN, Kafil HS, Yousefi M. Nanoparticles: Novel vehicles in treatment of Glioblastoma. Biomed Pharmacother. 2016 Feb;77:98-107. doi: 10.1016/j.biopha.2015.12.014. Epub 2015 Dec 29. PMID: 26796272). So, the authors need to outline a detailed examination of the challenges of GBM treatment and future prospectus of this review

Thank you for your suggestion for adding this important reference. Herein we provided a detailed examination of the challenges of glioblastoma (GBM) treatment and the future prospectus of this review in the section of the introduction. Moreover, this related reference has been cited as the ref. 20. Please kindly check with the revised introduction on page 7, lines 80-90, and reference 20 with the red mark.

Reference

20. Pourgholi, F.; Farhad, J.-N.; Kafil, H.S.; Yousefi, M. Nanoparticles: Novel vehicles in treatment of Glioblastoma. Biomed Pharmacother 2016, 77, 98-107.

2

In tables, please add the cellular pathways that are targetted through the treatment together with the references

Thank you for your kind concern. The treatment modalities related to the cellular pathways mostly involve the selected drug ontology. Therefore, we unified the cellular mechanistic description in Table 3. If the reference does not mention the cell therapy pathways, we express it as “n/a”. Please kindly refer to the revised Table 3 with red highlights.

3

Authors should include other types of recent combinational therapy with nano-particles such as radiotherapy (Pourgholi F, Hajivalili M, Farhad JN, Kafil HS, Yousefi M. Nanoparticles: Novel vehicles in treatment of Glioblastoma. Biomed Pharmacother. 2016 Feb;77:98-107. doi: 10.1016/j.biopha.2015.12.014. Epub 2015 Dec 29. PMID: 26796272); cold atmospheric plasma (He, Zhonglei et al. “Cold Atmospheric Plasma Stimulates Clathrin-Dependent Endocytosis to Repair Oxidised Membrane and Enhance Uptake of Nanomaterial in Glioblastoma Multiforme Cells.” Scientific reports vol. 10,1 6985. 24 Apr. 2020, doi:10.1038/s41598-020-63732-y); etc

Thank you for your comments and for sharing the important references with us. Based on the references, we intended to detail describe and cite those significant references in the new section: “4.3. Other strategies of GBM treatments”. Moreover, these related references have been cited as the ref. 20 and 101. Please kindly check the extra section of “4.3. Other strategies of GBM treatments” in the “4. Near-Infrared Phototherapy and Other Therapies of Brain Glioblastoma.” on page 11, lines 420-432.

References

20. Pourgholi, F.; Farhad, J.-N.; Kafil, H.S.; Yousefi, M. Nanoparticles: Novel vehicles in treatment of Glioblastoma. Biomed Pharmacother 2016, 77, 98-107.

101. Guerrero-Preston, R.; Ogawa, T.; Uemura, M.; Shumulinsky, G.; Valle, B.L.; Pirini, F.; Ravi, R.; Sidransky, D.; Keidar, M.; Trink, B. Cold atmospheric plasma treatment selectively targets head and neck squamous cell carcinoma cells. International journal of molecular medicine 2014, 34, 941-946.

4

I think, it will be interesting to mention about the dose (in concentration) of nanoparticles that are used in the studies in tables

Thank you for your kind suggestion. The information on adding the dosage of the drug will definitely bring detailed evidence for the GBM treatment. Thus, we added a new column to mention the dose of nanoparticles that are used in the references in Table 1, Table 2, and Table 3. Please kindly refer to the revised description with a red mark on those three tables.

Reviewer 2 Report

I would like to thank you for your efforts in providing this informative review on the development of various nanocomposites in diagnosing and treating GBM. this review can easily assist researchers in the field to design efficient injectable and implantable nanocomposites for brain cancer therapy. I believe that this review Is completely in line with this journal scope and can be published after the plagiarism checking and grammar checking process. 

Author Response

Response to reviewer 2:

No

Reviewer 2 comments

Changes to the manuscript

1

I would like to thank you for your efforts in providing this informative review on the development of various nanocomposites in diagnosing and treating GBM. this review can easily assist researchers in the field to design efficient injectable and implantable nanocomposites for brain cancer therapy. I believe that this review Is completely in line with this journal scope and can be published after the plagiarism checking and grammar checking process.

We thank the reviewer for your positive encouragement and kind suggestions. This review describes an approach to glioblastoma (GBM) theranostics that uses different nanoparticles. Hybrid nanoparticles can be applied as drug carriers to deliver anticancer drugs to the centers of GBM cells efficiently. Biocompatible hybrid nanoparticles have been synthesized for loading the different drugs to achieve cancer chemotherapy. Following intravenous injection, nanocomposites actively accumulate in the tumor site. They are subject to cross-reactive cellular uptake by cancer cells, resulting in augmented in vivo drug delivery enrichment in tumor cells.

Reviewer 3 Report

The general topic of nano materials for cancer treatment and imaging has been massively overviewed in many works, even in recent reviews. From the title I was expecting a focus on nanoparticles including different materials, that could provide independent functionalities.

However, many examples and descriptions are reporting single materials or independent drugs. Such parts should be profoundly reduced, and appropriate reviews should be cited. For example, the explanation of lanthanide-doped NPS, quantum dots… should be condensed and an effort should be given in describing on how such materials could be combined or assembled. The same should be done with the drugs (PTX, DOX, TMZ… and natural molecules). I would suggest that the authors emphases aspects of multimaterials or hierarchical systems containing different substances, that I consider that this would be the most innovative aspect of the review. I consider that the term composite in the title is the most appropriate: composite is more related to structures containing (usually a continuous and a discontinuous) phases consisting of materials often exhibiting distinct mechanical properties. Maybe multimaterials, for example, would be more appropriate.

Author Response

Response to reviewer 3:

No

Reviewer 3 comments

Changes to the manuscript

1

The general topic of nano materials for cancer treatment and imaging has been massively overviewed in many works, even in recent reviews. From the title I was expecting a focus on nanoparticles including different materials, that could provide independent functionalities. However, many examples and descriptions are reporting single materials or independent drugs. Such parts should be profoundly reduced, and appropriate reviews should be cited. For example, the explanation of lanthanide-doped NPS, quantum dots… should be condensed and an effort should be given in describing on how such materials could be combined or assembled. The same should be done with the drugs (PTX, DOX, TMZ… and natural molecules). I would suggest that the authors emphases aspects of multimaterials or hierarchical systems containing different substances, that I consider that this would be the most innovative aspect of the review.

Thank you for your comments. We have removed some single material or stand-alone drug narratives in order to avoid the possibility of defocusing the topic of glioblastoma (GBM) theranostics. We have shortened the explanation of single materials and drugs and added an introduction to composite materials and their advantages in paragraphs 3, 4, and 5. Meanwhile, several new related references have been cited to support the way in which multifunctional nanomaterials can be used for the theranostic of GBM. Please kindly check the revised sections of “3. Near-Infrared Imaging of Brain Glioblastoma” and “Nanoparticles Combined with Chemotherapy Applied in Curing Glioma.” and the new citations as follows: 20, 101, 117, 131, 179, and 180.

References

20. Pourgholi, F.; Farhad, J.-N.; Kafil, H.S.; Yousefi, M. Nanoparticles: Novel vehicles in treatment of Glioblastoma. Biomed Pharmacother 2016, 77, 98-107.

101. Guerrero-Preston, R.; Ogawa, T.; Uemura, M.; Shumulinsky, G.; Valle, B.L.; Pirini, F.; Ravi, R.; Sidransky, D.; Keidar, M.; Trink, B. Cold atmospheric plasma treatment selectively targets head and neck squamous cell carcinoma cells. International journal of molecular medicine 2014, 34, 941-946.

117. Banerjee, I.; De, K.; Mukherjee, D.; Dey, G.; Chattopadhyay, S.; Mukherjee, M.; Mandal, M.; Bandyopadhyay, A.K.; Gupta, A.; Ganguly, S., et al. Paclitaxel-loaded solid lipid nanoparticles modified with Tyr-3-octreotide for enhanced anti-angiogenic and anti-glioma therapy. Acta Biomater 2016, 38, 69-81.

131. Ruan, S.; Yuan, M.; Zhang, L.; Hu, G.; Chen, J.; Cun, X.; Zhang, Q.; Yang, Y.; He, Q.; Gao, H. Tumor microenvironment sensitive doxorubicin delivery and release to glioma using angiopep-2 decorated gold nanoparticles. Biomaterials 2015, 37, 425-435.

179. Li, Z.-Y.; Liu, Y.; Wang, X.-Q.; Liu, L.-H.; Hu, J.-J.; Luo, G.-F.; Chen, W.-H.; Rong, L.; Zhang, X.-Z. One-pot construction of functional mesoporous silica nanoparticles for the tumor-acidity-activated synergistic chemotherapy of glioblastoma. Acs Appl Mater Inter 2013, 5, 7995-8001.

180. Norouzi, M.; Yathindranath, V.; Thliveris, J.A.; Kopec, B.M.; Siahaan, T.J.; Miller, D.W. Doxorubicin-loaded iron oxide nanoparticles for glioblastoma therapy: A combinational approach for enhanced delivery of nanoparticles. Sci Rep-Uk 2020, 10, 1-18.

2

I consider that the term composite in the title is the most appropriate: composite is more related to structures containing (usually a continuous and a discontinuous) phases consisting of materials often exhibiting distinct mechanical properties. Maybe multimaterials, for example, would be more appropriate.

Thanks to the reviewer for the kind suggestions. Herein, we try to add more strategies of multi-materials in different sections. In paragraph 3, we propose imaging strategies to demonstrate the multiple imaging capability of the composites. In paragraph 4, other strategies of GBM treatments are proposed. Finally, we explained how the drugs and nanoparticles complex platform exhibits distinct mechanical properties after loading with different drugs in paragraph 5. Please kindly check the revised section of “3.6. Imaging Strategies” on page 9, lines 324-330, “4.3. Other strategies of GBM treatments” on page 11, lines 420-432, and “5.3. Drugs and Nanoparticles Complex Platform” on page 18, lines 691-730.
